# LoRa Technology Propagation Models for IoT Network Planning in the Amazon Regions

**DOI:** 10.3390/s24051621

**Published:** 2024-03-01

**Authors:** Wirlan G. Lima, Andreia V. R. Lopes, Caio M. M. Cardoso, Jasmine P. L. Araújo, Miércio C. A. Neto, Maria E. L. Tostes, Andréia A. Nascimento, Mauricio Rodriguez, Fabrício J. B. Barros

**Affiliations:** 1Computer and Telecommunications Laboratory (LCT), Institute of Technology (ITEC), Federal University of Pará (UFPA), Belém 66075-110, Brazil; caio.cardoso@itec.ufpa.br (C.M.M.C.); jasmine.araujo@gmail.com (J.P.L.A.); miercio@ufpa.br (M.C.A.N.); tostes@ufpa.br (M.E.L.T.); fjbbrito@gmail.com (F.J.B.B.); 2School of Electrical Engineering, Pontificia Universidad Católica de Valparaíso (PUCV), Valparaíso 2362804, Chile; andreia.rodrigues.l@mail.pucv.cl (A.V.R.L.); mauricio.rodriguez.g@pucv.cl (M.R.); 3Research and Development Program, Norte Energia S.A, Brasília 70714-900, Brazil; andreianascimento@norteenergiasa.com.br

**Keywords:** Amazon rivers, LoRa, uplink, IoT, LPWAN, path loss, dense vegetation

## Abstract

Designing and deploying telecommunications and broadcasting networks in the challenging terrain of the Amazon region pose significant obstacles due to its unique morphological characteristics. Within low-power wide-area networks (LPWANs), this research study introduces a comprehensive approach to modeling large-scale propagation loss channels specific to the LoRaWAN protocol operating at 915 MHz. The objective of this study is to facilitate the planning of Internet of Things (IoT) networks in riverside communities while accounting for the mobility of end nodes. We conducted extensive measurement campaigns along the banks of Universidade Federal do Pará, capturing received signal strength indication (RSSI), signal-to-noise ratio (SNR), and geolocated point data across various spreading factors. We fitted the empirical close-in (CI) and floating intercept (FI) propagation models for uplink path loss prediction and compared them with the Okumura–Hata model. We also present a new model for path loss with dense vegetation. Furthermore, we calculated received packet rate statistics between communication links to assess channel quality for the LoRa physical layer (PHY). Remarkably, both CI and FI models exhibited similar behaviors, with the newly proposed model demonstrating enhanced accuracy in estimating radio loss within densely vegetated scenarios, boasting lower root mean square error (RMSE) values than the Okumura–Hata model, particularly for spreading factor 9 (SF9). The radius coverage threshold, accounting for node mobility, was 945 m. This comprehensive analysis contributes valuable insights for the effective deployment and optimization of LoRa-based IoT networks in the intricate environmental conditions of the Amazon region.

## 1. Introduction

The North Region of Brazil, known worldwide for its fauna and flora, is also widely recognized for its rivers, effluent, and water regimes. Its rivers are essential for the survival of its inhabitants, and they are commonly called “roads”, acting as one of the main modes of transportation for the mobility and flow of production and the development of the economy [1].

The city of Belém, one of the few macro-regions of the North Region, is bathed by Guajará Bay, a meeting point between the mouth of the Acará River and the mouth of the Guamá River. Geographically, the city also has islands bathed by the bay, inhabited by riverside communities, where river transport is the only viable means of transportation, whether for the displacement of people or the flow and movement of the raw material, industrial, and food sectors [2]. Small- and medium-sized boats move along the river banks and serve the riverside communities in these regions. Telecommunications systems do not exist or are highly precarious because they are challenging mobile communication and broadcasting environments [3].

These regions need to catch up with the connectivity demands and enjoy the benefits of the new era of the IoT, which is expected to move about USD 19 trillion worldwide. Of this amount, USD 860 billion will specifically impact Latin American economies, and 40% (USD 352 billion) will be in the Brazilian IoT market alone [4].

The big challenge is determining the best IoT connectivity solution, and LoRa technology appears to be an alternative in several sectors for its long range, low power, and low cost. It is suitable for IoT applications that require secure and long-range communication, such as agriculture, smart cities, industry, and logistics [5]. Since it is wireless communication, such communication is given by links between the first and second layers, either between end nodes or between end node and gateway [6]. The data exchange from the gateway to the end node is downlink, and the data exchange from the end node to the gateway is uplink.

Considering the communication of a boat on a coastal river in the Amazon region, there is always a densely wooded area close to it. When downlink communication is possible, uplink communication is not necessarily possible, disrespecting the principle of reciprocity in LoRa applications [7]. Therefore, this work considers the uplink situation, representing the worst communication scenario among IoT devices.

LoRa-based wireless information exchange is an application for many purposes [8]. Monitoring, tracking, and locating objects are applications of LoRaWAN technology in maritime environments. Whether monitoring natural disasters, water quality, or vessels, the technology delivers significant results in cost-effectiveness and real-time collection [9,10,11,12,13,14,15,16,17,18,19,20,21,22,23,24].

Regarding water environments, measurements and channel modeling through LoRa technology are most commonly performed in maritime environments [9,22]. Few studies present channel modeling, coverage, or packet reception efficiency results in riverine environments [23,24]. However, the rivers in those studies have different characteristics from Amazonian rivers, which do not feature dense vegetation on their banks.

Considering the significance of rivers for the Amazon region, it is essential to include the communities living on their banks, while also considering the dense vegetation in these environments. Seeking an application of LoRa sensor networks that collect data present in a boat that travels on the banks of these rivers, we carried out extensive measurement campaigns on the edge of Universidade Federal do Pará (UFPA), bathed by the Guamá River. We fixed the LoRa gateway on the roof of the largest UFPA building, called Mirante do Rio, and used an end node in an unmanned aerial vehicle (UAV) of the drone type, moving for 1.4 km on a predetermined route, maintaining a distance of 50 m from the river bank. Our study presents results of signal coverage with mobility between the gateway and the LoRa end node at 915 MHz, packet reception rate data to quantify how viable the technology is for these types of environments, and the characterization of the channel when considering an LoRa device on a route on the banks of rivers. We analyze the sending of information from the end node to the network (uplink), examining the received values of RSSI and SNR using different spreading factor values. Additionally, we characterize the channel by adjusting models and compare them with the Okumura–Hata suburban model; finally, we propose a new uplink model that considers the vegetation loss present at ITU Recommendation P-833.

In this context, the contributions of this work are as follows:Extensive measurement campaigns in coastal environments of rivers;Characterization of uplink channels, for three different spreading factors (SF) values, to analyze which SF presents the best results for the environment under study;Adjustment of close-in (CI) propagation and floating intercept (FI) models, presenting path loss exponent (PLE) values for the uplink channel;Analysis of received signal strength indication (RSSI) and signal-to-noise ratio (SNR) variation along the route for the uplink channel;Analysis of the rate of received packets;Comparing adjusted models to classic Okumura–Hata model for suburban environments;Uplink channel modeling that considers the attenuation suffered by vegetation, based on the free space model and the ITU-R Recommendation P-833 model;Statistical analysis of coverage for LoRa network planning in river cases.

From our results, we aim to assist in the planning of LoRa sensor network links in regions along the rivers, and quantitatively verify the signal attenuation resulting from the morphology of the environment and dense vegetation present in the Amazonian coastal zones. Additionally, we will validate the use of technology for sending information securely, and quickly, even with the boat in motion, and support the development of new LoRa applications for use in smart campuses.

The rest of this article is organized as follows: Section 2 provides an overview of LoRa technology. Section 3 describes the equipment, the characteristics of the test environment, and the methodology used in the measurement campaigns. Section 4 describes all empirical propagation models used in the present study. Section 5 presents the measured results and the relevant work analysis. Finally, Section 6 presents the conclusions.

## 2. LoRa Overview

LoRa is a technology belonging to the physical layer of LPWAN (low-power wide-area network) technology, featuring wireless modulation that offers an attractive combination of long-range capability, low energy consumption, and security in data transmission [5,25]. Created by Semtech and promoted by the LoRa Alliance, LoRa is the physical layer that establishes the long-range communication link. LoRaWAN is the communication protocol based on the ALOHA [26] communication system architecture. This protocol and the star-type architecture influence the battery life, network capacity, security, and quality of service of applications connected to the network. The LoRa network architecture comprises the end devices or end nodes, gateways, network servers, and application servers. Figure 1 shows the LoRa network architecture.

End devices communicate with LoRa gateways via the LoRaWAN protocol. Gateways serve as bidirectional packet converters, transmitting packets from end devices to the network and vice versa. The communication between end devices and the network involves uplink and downlink. Transmitting packets from end devices to the network is the uplink message, and sending packets from the network to end devices is the downlink message.

The LoRaWAN protocol is responsible for defining the communication and architecture of a system specifically designed for network sensors. It employs low data rates and extended time for communication, impacting network capacity, node lifetime determination, service quality, security, and various connected applications.

This protocol facilitates communication between a large chain of peripheral devices and a single gateway by enabling medium access control (MAC). However, the end nodes are not tied to a specific gateway, which may lead to data loss. To mitigate this, multiple gateways can receive the data, thereby increasing the range of communication between devices and the gateway.

For the network layers of LoRa technology and the LoRaWAN protocol, the definition is generally structured into three layers: the physical layer, the data link layer, and the application layer. The LoRa layer combines the physical layer (wireless communication and signal modulation) with the radio frequency layer, containing the frequency bands the technology operates in. The data link layer encompasses the LoRaWAN protocol classes and the LoRaWAN protocol itself, while the application layer represents all the end applications that can be realized using the LoRaWAN protocol.

According to the LoRa Alliance, communication between layers of LoRa technology is shown in Figure 2.

There are two layers of protection for data security in LoRa: the network layer and the application layer. Network layer security ensures the authenticity of the node in the network, while application layer security ensures that the network operator cannot access the end user’s application data.

LoRa end devices can be categorized into three classes: A, B, and C. These classes represent the trade-off between network downlink communication and battery life. Class A devices require the least amount of system power and support bidirectional communication, both uplink and downlink. Class B devices are also bidirectional, but the gateway can be programmed to receive specific signals. On the other hand, Class C devices have downlink latency and cannot send and receive packets simultaneously. While Class A is fundamental, Classes B and C are optional. In this study, we utilized LoRa end device Class B [25].

Due to its unique characteristics and capabilities, LoRa technology has great potential for scientific research and a wide range of applications. Key points to consider are the following:Long-range, low-power communication: LoRa’s spread spectrum modulation and chirp spread spectrum techniques enable communication over long distances, making it ideal for scenarios where low power consumption is crucial, making it suitable for applications like IoT.Versatile data rates and payloads: LoRa provides flexibility to accommodate diverse application requirements through adaptive data rates and varying payload sizes.Unlicensed spectrum usage: Operating in unlicensed ISM bands (e.g., 868 MHz, 915 MHz) enables LoRa to bypass the complexities and costs associated with licensed spectrum, facilitating widespread adoption.Low-cost infrastructure: LoRa networks, such as LoRaWAN, require minimal infrastructure investment, making it cost-effective for both research projects and large-scale deployments.Security measures: Robust security features, including encryption and authentication, make LoRa suitable for secure data transmission in applications like healthcare and critical infrastructure monitoring.Diverse applications: LoRa technology finds applications in various sectors, such as agriculture, environmental monitoring, smart cities, and asset tracking, contributing to data-driven scientific research and improving operational efficiency.

Table 1 presents some related works on LoRa applications over water and in environments with vegetation. It provides information on the operating frequency of LoRa technology, the type of condition (line-of-sight (LOS) and/or non-line-of-sight (NLOS)), the study scenario, coverage, and the spreading factor under investigation.

## 3. Measurement Methodology

This section presents a detailed description of the scenario, the equipment, the measurement setup, and the experimental procedure used in our measurement campaign.

### 3.1. Measurement Methodology

We conducted measurement campaigns on the shores of UFPA along the Guamá River. An LoRa gateway was placed on the roof of the Mirante do Rio building, 30 m above the ground. Simultaneously, an LoRa device was mounted on a drone, flying over the riverbanks along the edge of UFPA. We calculated the Euclidean distance between the transceivers using the latitude and longitude values captured by the GPS module in the drone’s LoRa device. The mobile end node stored the GPS position in motion to collect uplink channel information and transmitted it in the LoRa packet to the gateway (uplink).

We utilized an end node consisting of an Arduino UNO board and a Dragino-developed shield plate, equipped with LoRa SX1276 and Quectel L80 GPS modules. The device was powered by a 5V USB and connected to a LoRa gateway, specifically the RAK7249 model. The end node was attached to a drone, running the LoRaWAN protocol as a transmitter. Figure 3 and Figure 4 show the end node and the gateway used in the measurement campaign. The data collected at the gateway were stored in LoRa packets and transmitted to the LoRa network server. The LoRa gateway RAK7249 has LoRaWAN public settings, allowing it to receive different spreading factors (SFs) and various bandwidths, with a sensitivity of −148 dBm.

The LoRa gateway employs Message Queuing Telemetry Transport (MQTT), a messaging protocol designed for low-bandwidth networks and IoT devices with high latency, to manage an internal server [35]. This internal server, in conjunction with the LoRaWAN protocol, enables the gateway to receive data in JSON format and store persistent data through an MQTT broker server. We utilized a local broker server to receive the LoRa frame data and store them in a relational database for future reference. Upon receiving the LoRa frame, the gateway provides crucial information such as RSSI, SNR, the end node address in HEX, and the payload message. The local server then records these parameters for future analysis and processing.

The end node transmitter attached to the drone sends packets containing GPS coordinates every second without using control mechanisms or automatic retransmissions. The radiated power is set to 20 dBm, employing SF7, SF8, and SF9 to analyze receiver sensitivity variation, with a code rate of 4/5 and a bandwidth (BW) set to 125 kHz on the 915 MHz channel. The packet payload is 40 bytes, including a 13-byte MAC header in the case of configurations. Table 2 shows the measurement configuration parameters.

We positioned the gateway on the terrace of the Mirante do Rio building, 30 m above the ground. The end node was then moved along the route at an average speed of 25 km per hour, fixed on the drone, and positioned 3 m above the ground. This methodology allows for the measurement of signal attenuation through RSSI values and the evaluation of SNR in motion for the uplink channel.

### 3.2. Measurement Scenario

The objective of this work is to characterize the channel for LoRa technology in coastal environments to facilitate the implementation of telecommunication networks serving the riverside communities of the Amazon region.

The chosen measurement scenario is the edge of UFPA, featuring a predefined route along the banks of the Guamá River. The area includes dense and tall vegetation, hindering the line of sight in many parts, along with medium-height buildings and open spaces used as parking lots. The study route spans 1461 m along the campus edge, extending from the Basic sector to the Health sector. Using the Google Earth tool, Figure 5 shows a satellite image of the measurement scenario.

Observing from the heights of the gateway and the end node, it is evident that the majority of the route is obstructed by vegetation and some buildings, as depicted in Figure 5. The portion obstructed by vegetation constitutes 1133 m, equivalent to 77% of the total route.

As mentioned in Section 2, uplink communication occurs between the device and the gateway, and the received data are then considered for calculating the RSSI, SNR, and packet efficiency at the gateway. Additionally, GPS data are collected to determine the locations of the measured points.

Due to the high battery consumption of the drone, we repeated the procedure four times, resulting in a total of 497, 437, and 461 points collected for SF7, SF8, and SF9, respectively. We captured 1395 sets of raw RSSI and SNR data for uplink. The signal exhibited similar behavior to the distance; on average, only four measurements were sufficient.

The motivation for choosing this environment stems from the significant importance of rivers in the Amazon region for transportation and the economy. By employing sensing with LoRa technology, we can locate a boat along the shores of UFPA Orla and obtain crucial data from boats that follow a similar route. These features underscore the importance of studying LoRa performance and conducting large-scale channel characterization in environments with extensive vegetation morphology, rivers, and suburban areas.

We adopted an average drone speed of 25 km/h, repeating the measurements four times along the route. Considering the average speed that a small vessel maintains for hours, it becomes evident that the ideal cruising speed falls within the range of 15 to 26 knots, corresponding to approximately 27.78 km/h to 48.15 km/h [36]. The distance between the gateway and the end node was determined by GPS coordinates, and we adopted an average distance of 50 m between the drone and the UFPA coast along the entire route. The route is shown in Figure 6.

### 3.3. Data Processing

After each measurement along the route, the data sent to the LoRa gateway were stored and transmitted to an MQTT broker server. Received signal strength indicator (RSSI), signal-to-noise ratio (SNR), spreading factor, and GPS position data were downloaded in .tx files and processed in codes using the MATLAB© R2023a computational tool. With the data already loaded into MATLAB, we focused on the dataset containing all the necessary information for modeling. RSSI and SNR data are the parameters used to calculate the path loss value based on [29,37], as outlined in:(1)$$PL=\left\{\begin{array}{cc} Pt+Gt+Gr-RSSI, & SNR>0\\ Pt+Gt+Gr-(RSSI+SNR), & other \end{array}\right.$$
where PL is the path loss in dB, Pt is the transmitter power, and Gt and Gr are the gains of the transmitting and receiving antennas, respectively. We modeled the data to characterize the uplink channel, by adjusting the classic close-in and floating intercept models. The adjusted parameters include the path loss exponent (*n* or PLE), the α intercept, and the β slope for spreading factors 7, 8, and 9, respectively. We compared fitted models with Okumura–Hata for suburban environments. Additionally, we propose a model for characterizing the vegetation attenuation based on Recommendation ITU-R P-833. Introducing this new model factor, we considered the Friis free space formula and the attenuation model for channel characterization in environments with dense vegetation typical in the Amazon region. We created a MATLAB© code to model and adjust all propagation models for the uplink channel.

## 4. Propagation Model

In this section, we will explore commonly used propagation models in LoRaWAN, including the close-in, floating intercept, and Okumura–Hata models. The study of wave propagation involves analyzing the distribution of energy in a signal through space, and propagation models aid in predicting the received signal power in a wireless network. By considering factors such as the morphology of the environment, terrain morphology, operating frequency, and the height of the transmitting and receiving antennas, we can calculate values and make adjustments for the propagation models [26,38].

### 4.1. Close-In Model

The close-in model estimates path loss both indoors and outdoors, with the path loss exponent as its adjustment parameter. Utilizing minimum mean square error (MMSE), the CI model employs linear regression to minimize error and standard deviation [37]. It is grounded in the free space model, using a reference distance as a physical basis that characterizes the path loss dependence between the transmitter and the receiver. In this study, we adopted a reference distance value of 1 m [38]. The CI model equation is given by:(2)$$PL^{CI}\left(f,d\right)=FSPL\left(f,d_{0}\right)+10n\log_{10}\left(\frac{d}{d_{0}}\right)+X_{\sigma }^{CI}$$
(3)$$FSPL=10\log_{10}{\left(\frac{4\pi d_{0}}{\lambda }\right)}^{2}$$
where FSPL is the initial loss in dB and represents the anchor point of the model, *n* is the loss exponent, *d* is the distance to estimate the loss amount, $d_{0}$ is the reference distance, which is 1 m, $X_{\sigma }^{CI}$ is a random variable with zero mean Gaussian distribution, and standard deviation σ also in dB, and λ is the wavelength as a function of frequency [38].

### 4.2. Floating Intercept Model

The FI model, also known as the alpha-beta model, is widely used in the Third Partnership Project (3GPP) and WINNER II standards [39,40,41,42]. Unlike the CI model, the FI model does not have a physical anchor, relying solely on the trajectory of the measured data to adjust the intercept (α) and the slope (β) of the loss curve through the MMSE method. The FI model equation is given by:(4)$$PL^{FI}\left(f,d\right)=\alpha +10\beta \log_{10}\left(\frac{d}{d_{0}}\right)+X_{\sigma }^{FI}$$

The path loss in dB, the floating intercept, and the model slope are represented by $PL^{FI}\left(f,d\right)$, α, and β, respectively. Large-scale fluctuations in average path loss with distance caused by zero-mean Gaussian shading can be represented by the random variable $X_{FI}^{\sigma }$ in dB. The values of α and β in the FI model are similar to the FSPL and PLE of the CI model, respectively. Although the FI model can be used in mmWave bands, 3GPP employs it for bands below 6 GHz, as stated in [42].

### 4.3. Okumura–Hata Model

The Okumura–Hata model aligns with Okumura’s original results as described by Hata. Hata’s expressions are tailored for communication in urban areas with flat topography and are applicable within the frequency range of 150 and 1500 MHz [43]. The Okumura–Hata model for urban environments is presented:(5)$$\begin{array}{c} L_{urb}=69.55+26.16\log_{10}f-13.82\log_{10}h_{t}-\\ A\left(h_{r}\right)+\left(44.9-6.55\log_{10}h_{t}\right)\log_{10}d \end{array}$$

The frequency *f* is in MHz, the heights of the transmitting antennas $h_{t}$ and receiving antennas $h_{r}$ are in meters, the correction factor $A(h_{r})$ due to the environment (urban or rural) is in dB, and the distance between antennas *d* is in km.

The $A(h_{r})$ value varies based on the analyzed environment, with each environment having a specific calculating method for the factor. This approach ensures an accurate and realistic modeling of the scenario under study. For the environment in this study, the factor calculation considers medium and small cities, as follows:(6)$$A\left(h_{r}\right)=1.1\log_{10}f-0.7)h_{r}-\left(1.56\log_{10}f-0.8\right)$$

Considering that the UFPA environment is characterized as suburban, we modified the Okumura–Hata model [44]. Therefore, the path loss can be represented by:(7)$$L_{suburb}=L_{urb}-2log_{10}{\left(f/28\right)}^{2}-5.4$$

Loss models based on large-scale measurements in a wireless channel provide realistic channel characterization [45]. Phenomena such as reflection, diffraction, refraction, and scattering are associated with the propagation behavior of electromagnetic waves, altering the wave’s behavior and introducing randomness in the transmitted wave, affecting its reception [29,38]. As mentioned in the previous section, we adopt the modeling approach of [29,37], which uses RSSI and SNR values to calculate path loss.

### 4.4. Recommendation ITU P-833 and the Vegetation Loss Model

Figure 5 and Figure 6 depict significant vegetation between the gateway and LoRa end node. In this context, we present previous studies that examine the attenuation caused by vegetation loss [31,46,47,48]. In [31], they consider two parameters related to trees—the leaf area index and the diameter of the tree trunk—for propagation loss studies in the LoRa channel, but they do not provide results for a link with dense vegetation. In [46], the LoRa operating frequency is 433 MHz and investigates the attenuation produced by the leaves. The study in [47] explores the transmission capacity of LoRa technology in environments covered by vegetation, characterized by attenuation due to foliage, branches, and tree trunks. The study involves multiple LoRa devices connected to the gateway, with a maximum distance of 200 m. Reference [48] highlights possible factors responsible for degrading the quality of the LoRa channel through the leaf medium, but it does not analyze dense vegetation.

Therefore, studying the effect of Amazon vegetation on the LoRa uplink channel is crucial. ITU Recommendation P-833-10 provides several models for evaluating the impact of vegetation on radio wave signals, applicable to various path geometries and vegetation types [49]. The Recommendation also includes measured data on vegetation fading dynamics and propagation delay characteristics. The Recommendation presents the additional loss due to vegetation, and it can be characterized based on two parameters:The specific attenuation rate (dB/m) refers to the reduction in signal strength per unit distance due to energy dispersion occurring outside the direct path between the transmitter and the receiver;The maximum additional attenuation due to vegetation in a radio path (limited by other mechanisms) is the sum of propagation of surface waves over the top of the vegetation medium and direct scattering within it (in dB).

Figure 7 illustrates the radio path in a woodland area, with antennas positioned outside the woodland. Key factors influencing signal quality include the elevation angle (θ) at which the radio signal travels, the length of vegetation (*d*) along the signal path, the average height of trees ($h_{v}$), the height of the receiving antenna above the ground ($h_{a}$), and the distance between the antenna and the roadside forest ($d_{w}$). In the modeling link shown in Figure 7, the transmitting equipment is higher than the receiving equipment. However, in our measurement methodology, the receiving equipment is higher than the transmitting equipment. This is because communication between LoRa devices is of the uplink type; the message is sent from the end node on the drone and received at the gateway fixed at Mirante do Rio. Consequently, our analysis considers an elevation angle value ($\theta_{d}$) dependent on the total vegetation depth distance ($d_{vegetation}$).

Figure 8 illustrates the location of the LoRa gateway (blue icon), the LoRa end node (with an icon), and the vegetation depth (red rectangle) used to calculate the elevation angles ($\theta_{d}$). These elevation angles depend on the depth of vegetation along the entire route. The orange icons represent the locations where data were measured, as shown in Figure 6. It is important to note that we used total vegetation depth values throughout the route. This implies that for each SF value, we collected various distance values between the gateway and the LoRa end node, and then added each distance value with vegetation between the devices to calculate the elevation angles.

The Recommendation presents modeling for vegetation in different seasons. Considering the seasonal characteristics and different vegetation of the Amazon region, we applied the following vegetation attenuation model:(8)$$Veg_{loss}\left(dB\right)=A*f^{B}*log10\left(d_{vegetation}\right)*{\left(\theta_{d}+E\right)}^{G}-4$$

Factor B represents the frequency-dependent attenuation due to seasonal characteristics and is given by:(9)$$\begin{array}{c} B=\left(0.30281-0.003624*kh\right){\left(f/1000\right)}^{\left(0.0013118-0.026236*kh\right)} \end{array}$$
where *f* is the frequency (MHz), the distance $d_{vegetation}$ is the total distance of the vegetation depth (m), $\theta_{d}$ is the distance-dependent elevation angle (degrees), kh is the season factor, and in the Southern Hemisphere, kh=6−|Month−6.5|. *A*, *B*, *E*, and *G* are empirical parameters and are determined for each area. In this study, we used the Kenyan cedar values in Table 3 of the Recommendation due to their similarity with the trees in the scenario.

## 5. Results

This section presents our results in four subsections: Analysis of RSSI and SNR Variation and Package Efficiency by SF, Path Loss, the proposed model with Vegetation Attenuation, and Statistical analysis for Coverage Threshold.

The subsection “Analysis of RSSI and SNR Variation and Rate of Received Packets” illustrates the variation of RSSI and SNR values collected for the three SF values under study (7, 8, and 9) and analyzes the rate of received packets. The “Path Loss” section analysis for uplink channels presents two analyses: the path loss variation for the three SFs and the path loss in SF7, SF8, and SF9, comparing adjusted CI and FI models with the classic Okumura–Hata model. In the subsection “Uplink Channel Modeling considering Attenuation by Vegetation”, we present the modeling that includes vegetation attenuation and compare it with models fitted and with the classic Okumura–Hata. Finally, the subsection “Statistical Analysis for Coverage Threshold” introduces the concept of quartiles to estimate the coverage radius from a dataset of the uplink signal in the environment under study.

### 5.1. Analysis of RSSI and SNR Variation and Rate of Received Packets by SF

The en-route propagation analysis addresses the non-line-of-sight (NLOS) conditions arising from obstructions like trees and buildings between the transmitter (Tx) and receiver (Rx) antennas. These obstructions impact radio link propagation performance, and the spreading factor (SF) value also plays a crucial role through airtime and signal sensitivity. It is essential to analyze how the NLOS condition affects packet efficiency based on the spreading factor type and the variation in RSSI and SNR values measured along the route.

The box plot in Figure 9 illustrates the variation of RSSI values collected in SF7, SF8, and SF9. Across the route, values range from −60 to −110 dBm, with a maximum of −60 dBm for SF8 and −110 dBm for SF7 and SF8. It is noteworthy that sensitivity and SF type are directly proportional, meaning lower SF values result in lower signal sensitivity. However, lower SF values also lead to shorter airtime. The RSSI data range for SF7, SF8, and SF9 is from −64 to −110 dBm, −60 to −110 dBm, and −65 to −115 dBm, respectively. The average RSSI value for SF7, SF8, and SF9 was −84.62 dBm, −85.85 dBm, and −87.76 dBm, respectively.

The box plot in Figure 10 displays the variation of SNR values collected in SF7, SF8, and SF9. The signal-to-noise ratio (SNR) measures the strength of a signal to background noise. The noise floor usually sets the limit for detecting a signal; however, LoRa technology can operate below the noise floor and decode signals at SNR values as low dB−20 dB [50]. SNR values in LoRa typically range between −20 and 10 dB. A value close to 10 dB indicates a stronger received signal and less noise.

In our results, SNR data range from 10.8 to −1.8 dB, 12.5 to −5.5 dB, and 13.8 to −9 dB for SF7, SF8, and SF9, respectively. The average SNR value was 7.54, 8.71, and 8.76 dB for SF7, SF8 and SF9, respectively.

RSSI and SNR values are key indicators of the capabilities of an LoRa radio receiver’s capabilities. RSSI measures the signal level, while SNR determines the quality of the received signal, indicating how close it is to failure or noise [25]. Table 3 displays the dispersion grade values, represented by the standard deviation (σ) in dB for RSSI and SNR for SF7, SF8, and SF9.

In analyzing the rate of received packets, it is important to note that some data collected at the gateway may lack RSSI and SNR information. To address this, we implemented a counter at the gateway that tracks the amount of data with and without RSSI and SNR information. This allows us to determine the rate of received packets by calculating the percentage of data containing RSSI and SNR information compared to the total amount of data sent and collected.

For analyzing the percentage of packets received, we use the ratio between the total number of packets sent (PS) and the total number of packets received (PR) to calculate the rate of received packets (RRP):(10)RRP(%)=PR/PS

The rate of received packets (RRP) depends on the signal level and quality (noise level). RRP values are 75.78%, 86.29%, and 92% for SF7, SF8 and SF9, respectively. These values are supported by the highest SNR values for SF9, SF8, and SF7, respectively. Average RSSI and SNR values further validate RRP, emphasizing the SF9 data with the highest RRP compared to the other SF values under study. Table 4 shows the average RSSI and SNR values and the RRP values for the three SFs.

### 5.2. Path Loss

The fitting model is a powerful tool for estimating attenuation in a real environment. We employed the CI and FI models to analyze uplink data, comparing them with the Okumura–Hata model. These models and parameters enable the estimation of attenuation in environments similar to the one studied in this work.

We will present our path loss values in two ways: first, by showing the path loss variation for the SF under study; then, by separating the modeling results for the three SFs.

#### 5.2.1. Path Loss Variation

We use Equation (Equation 1) to calculate the path loss from measured RSSI and SNR data. As seen in Figure 9 and Figure 10, each SF presents values with different ranges, reflecting in the path loss values, particularly when the range of SNR values includes positive values above 10 dB and negative values below −5 dB. Figure 11 illustrates that SF9 exhibits the highest path loss values, reaching a maximum of 142.3 dB, while SF8 shows a minimum of 80 dB. The obstruction caused by vegetation between the end node and the gateway contributed to a higher path loss value. Despite this obstruction, the SF9 values can cover a greater distance, encompassing the entire established route, due to having more airtime.

Maximum values were 130 dB, 132.5 dB, and 142.3 dB; minimum values were 84 dB, 80 dB, and 85 dB for SF7, SF8, and SF9, respectively. The average values were 104.6 dB, 105.9 dB, and 107.9 dB for SF7, SF8 and SF9, respectively. The standard deviation values (σ) are 10.10, 10.43, and 11.12 for SF7, SF8, and SF9.

#### 5.2.2. Path Loss for Uplink Channels for Spreading Factors 7, 8, and 9

We present graphs depicting data collected in SF7, SF8, and SF9 along the route. We modeled the data by adjusting the CI and FI models and comparing them with the Okumura–Hata model for suburban environments. The anchoring value of the CI model for the frequency of 915 MHz is 31.67 dB, and this value will be used to model the three SFs under study.

As mentioned before in Section 4, the *n*, α, and β parameters are adjusted through the MMSE method, resulting in *n* = 2.69, α = 29.70, and β = 2.76, for SF7. Notably, in Figure 12, the two adjusted models present similar behavior, as the values of β and *n* are very close. The uplink channel experiences a more significant loss with distance in the measurement path, and the *n* value obtained is greater than the free space conditions due to the high density of trees on the route. The precision of the models is evaluated with a root mean square error (RMSE) between the data and models, yielding values of 6.17 dB, 6.17 dB, and 6.91 dB for CI, FI, and Okumura–Hata, respectively. As the values of *n* and β are close, the values of RMSE for the CI and FI models are also close.

For SF8, as shown in Figure 13, the values are n=2.73, α=26.19, and β=2.93. Due to the sensitivity of the signal in SF8, its average loss value is greater than the average signal loss value in SF7, resulting in regressions in CI and FI with different slopes, as observed in the graphic. Since there is no significant difference in the slope values of the CI and FI models, it can be concluded that the difference between the RMSE values will be negligible. In SF8, the RMSE values for the CI, FI, and Okumura–Hata models are 6.57 dB, 6.55 dB, and 7.53 dB, respectively.

Figure 14 shows that the SF9 has values of n=2.78, α=19.16, and β=3.23. Due to the higher average path loss value compared to SF7 and SF8, SF9 presents a significant difference in modeling with CI and FI. This is because SF9 has greater sensitivity to the RSSI and SNR values.

In contrast to the observations made in the modeling with SF7 and SF8, the modeling with SF9 revealed slope values with significant differences, which will be reflected in the RMSE values. For SF9, the RMSE values are 6.68 dB, 6.57 dB, and 7.87 dB, respectively, for the CI, FI, and Okumura–Hata models. Our study found that the FI model had the best RMSE values among all SFs. The CI model had the second-best RMSE value, and the Okumura–Hata model had the highest RMSE value.

Several articles in the literature focus on determining the maximum range distance of LoRa technology for links, presenting values of 6 km for urban and suburban environments, 18 km for rural environments [27,29,30,31,51,52], and between 0.4 and 30 km for environments over water [9,17,18,19,27]. Our experiment was conducted on a route located 50 m away from the riverbank in an environment different from the environments studied in the literature. Our aim is not to determine the coverage or range of LoRa technology across a river, but, rather, to characterize the signal on the banks of the river. We conducted our test on a route that connects two campuses within the university, and our results are applicable for links between distances of 1 to 2 km. These results can be used to account for dense vegetation on the path, as there are many islands in the rivers of the Amazon.

### 5.3. Uplink Channel Modeling Considering Attenuation by Vegetation

The CI and FI models provide more accurate adjustments than the Okumura model and help plan IoT networks in environments similar to the study environment. However, adjusting the CI and FI model parameters is impossible if measurement campaigns cannot be performed in the study environments. Furthermore, the Okumura–Hata model has a significantly higher RMSE value when compared to the RMSE values of the CI and FI models in all SFs under study. Our objective was to measure the reduction in signal strength caused by thick vegetation in our current setup. Additionally, we planned to investigate the impact of selecting different spreading factors (SF) on LoRa technology while taking into account the longer transmission time and higher sensitivity associated with higher SF values. Based on these analyses, we aimed to identify the most suitable SF for a reliable and robust link in this environment.

Therefore, we propose a propagation model to predict vegetation losses using the free space model added to the vegetation attenuation calculated through the models present in Recommendation ITU-R P-833 ($Veg_{loss}$). We use (Equation 8) to calculate all attenuation values based on all elevation angles, with an average tree height of 25 m, operating frequency of 915 MHz, A=1.5, B=0.28, E=0.01, G=−0.12, kh=2.5, and p=121.1. Thus, the proposed model is shown as follows:(11)$$PL_{veg}\left[dB\right]=20*log_{10}\left(\lambda /4\pi d\right)+Veg_{loss}$$
where *d* is the distance between Tx and Rx, λ is the wavelength as a function of frequency, $Veg_{loss}$ is the vegetation attenuation calculated through the models present in Recommendation ITU-R P-833, and $PL_{veg}$ is the attenuation vegetation model.

Figure 15, Figure 16 and Figure 17 show the modeling with the CI, FI, Okumura–Hata, and the proposed model $PL_{veg}$ models for SF7, SF8, and SF9, respectively.

Table 5 summarizes all parameters for all models presented in the present work.

Our study found that the FI model had the best RMSE values among all SFs, and the CI model had the second-best RMSE value. We can observe differences between theoretical data (represented by modeling with the Okumura–Hata (O–H) model) and empirical data (represented by modeling with the floating intercept (FI) model). This difference is accentuated by the fact that O–H modeling was proposed for static links; however, our measurement methodology involves a moving device (dynamic). Considering the mobility of the LoRa device, the attenuation caused by multipath and physical phenomena such as diffraction, dispersion, and shadowing affect the path loss value. This path loss, calculated with empirical data, is significantly different and greater than the loss calculated for a link with a static device when the device is motionless and collecting data. Considering that our device is moving and that there is dense vegetation between the LoRa device and the gateway, we observe an average difference between the RMSE values of the O–H model and the FI model of 0.74 dB, 0.98 dB, and 1.3 dB, for SF7, SF8, and SF9, respectively. According to Figure 15, Figure 16 and Figure 17, the FI (in blue) and O–H (in red) models show no significant difference in SF7 and SF8. However, in SF9, a more noticeable difference is observed due to the sensitivity of the signal and its attenuation caused by environmental vegetation. The O–H model is also featured in [29], where it was modeled in a rural environment with an RMSE value of 6.9 dB, using data measured in SF12 at an operating frequency of 868 MHz. The authors of that study also highlighted the substantial difference between empirical and theoretical data.

The proposed model for the uplink channel with dense vegetation yields RMSE results that vary depending on the studied SF. Specifically, the RMSE values for SF7 and SF8 are higher than those obtained from all the other models. In contrast, for SF9, the RMSE value is lower than the Okumura–Hata model’s RMSE value.

The maximum coverage or range of LoRa technology using SF12 in environments with dense vegetation is 2 km, while the minimum is 250 m, according to studies [28,32,33,34,53,54,55].

Our study demonstrates that SF9 provides reliable coverage of up to 1.5 km when considering dense vegetation and links over water, with received packet rates exceeding 90%. In contrast, ref. [9] reports a packet reception rate drop to 68% over water for SF12.

There is a lack of analyses on the rate of received packets in the literature. In [30], they examined the packet delivery rate at a speed of 100 m/min with SF7, SF9, and SF12, reporting a rate of approximately 30% for SF9 at a distance of 200 m.

Increasing the SF in LoRa technology enhances the range and airtime of the signal. However, this also increases the signal’s sensitivity to multipath and obstructions between LoRa devices. Our objective is to investigate whether a higher SF would result in a greater range in our scenario, which involves dense vegetation. Additionally, we aim to determine if modeling vegetation would yield lower root mean square error (RMSE) values. Based on our results, we confirm that even with dense vegetation and increased sensitivity, SF9 demonstrates the best performance in our study scenario.

LoRa link operators can use a model that considers vegetation, SF9, tree height, device height, and distance between devices in applications on riverbanks and with dense vegetation.

### 5.4. Statistical Analysis for Coverage Threshold

To analyze the coverage radius threshold, we adopt the statistical approach introduced in [56], which uses the concept of quartiles to estimate the coverage radius from a dataset. Observations arranged in ascending order are divided into four equal parts to obtain quartiles. The first quartile (Q1) comprises the initial 25% of the data. The second quartile (Q2) is the median, representing the range between 25% and 50% of the data. The third quartile (Q3) spans the range from 50% and 75%, while the fourth quartile (Q4) covers 75% and 100% of the data. In the context of determining the limit coverage radius for a moving end node that sends data to the gateway (uplink communication), we focus on Q3.

Figure 18 shows the representative graphic for the 75th percentile for SF7, SF8, and SF9.

The coverage radius threshold was established based on the 75th percentile values of SF7, SF8, and SF9, ensuring a reliable communication link between mobile objects uplinking data to end nodes and the gateway. These values were measured at 954.3 m, 905.3 m, and 975.8 m, respectively.

Our analysis focuses on dynamic network links, where the final device is in motion, simulating a boat with a speed of approximately 25 km/h. The Q3 values represent the coverage radius considering continuous information transmission and reception at the LoRa gateway.

For values below Q3, the coverage radius is conservative. For distance radius values above the Q3 values, the information sent in uplink would still be received by the gateway but not continuously. In all cases of SFs measured, there was a connection between the end node and the gateway along the entire route, with information sent confirmed by its RRP values.

## 6. Conclusions

The central objective of this study was to characterize the uplink LoRa channel at 915 MHz to meet the IoT connectivity demands in Amazon rivers. The combination of long-range communication, low power consumption, security features, and affordability in LoRa technology makes it a valuable tool for scientific research and a catalyst for innovation in a wide array of IoT applications. In this context, understanding the propagation loss in the study environment is essential for developing a realistic channel characterization.

We conducted extensive measurement campaigns on the banks of the river at the Federal University of Pará. We installed an LoRa gateway on the roof of a building and deployed the end node on a drone to collect received signal strength indication and signal-to-noise ratio values, along with geolocated points. The measurements were taken at three different spreading factors (SF7, SF8, and SF9) along a predetermined route.

Additionally, we applied the close-in (CI) and floating intercept (FI) models to estimate the path loss along the study route and compared them with the Okumura–Hata model. Moreover, we introduced a novel propagation model for dense vegetation environments ($PL_{veg}$), utilizing a vegetation attenuation model from ITU-R P-833 and the Friis free space formula. This model can be employed to estimate path loss values in areas where direct measurements are not feasible. The root mean square (RMS) error values were 6.17 dB, 6.17 dB, 6.91 dB, and 8.43 B, for the CI, FI, Okumura–Hata, and $PL_{veg}$ at SF7; 6.57 dB, 6.55 dB, 7.53 dB, and 7.98 dB for CI, FI, Okumura–Hata, and $PL_{veg}$ at SF8; and 6.68 dB, 6.57 dB, 7.87 dB, and 7.5 dB, for CI, FI, Okumura–Hata, and $PL_{veg}$ at SF9.

We encountered several limitations during our study, including challenges in controlling the drone over water due to reflections on the camera attached to it, the absence of works involving mixed environments and water, the limited battery life of the drone (approximately 20 min), the region’s Amazon climate characterized by rapid and intense rains during the summer, which posed a risk to our measuring equipment, and the size of the route, covering the territorial limits of the university. The combination of these factors, along with the drone’s battery constraints, made it impractical for us to conduct additional routes over water.

Despite these limitations, our study yielded promising results. This paper contributes models for uplink planning in LoRa networks for monitoring boats or smaller vessels. The analysis of the receiver packet rate demonstrates that the maximum packet received with mobility is 92.

Future work will encompass propagation studies to analyze high data throughput resulting from the use of two or more connected LoRa devices transmitting information to the LoRa gateway. This will include exploring spreading factors beyond 9 for comparison with results described in the literature (see Table 1). Additionally, we plan to conduct new routes over water, covering distances greater than 2 km, to further compare our results with those outlined in Table 1. Another aspect of future investigation involves varying the height of the LoRa device to analyze path loss and its increased influence on vegetation.

## Figures and Tables

**Figure 1 sensors-24-01621-f001:**
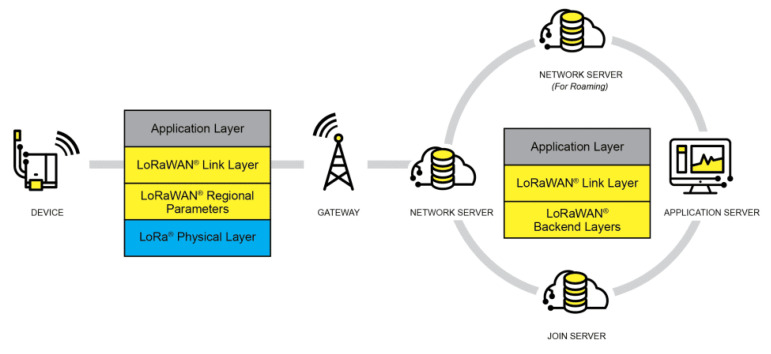
LoRa network architecture [25].

**Figure 2 sensors-24-01621-f002:**
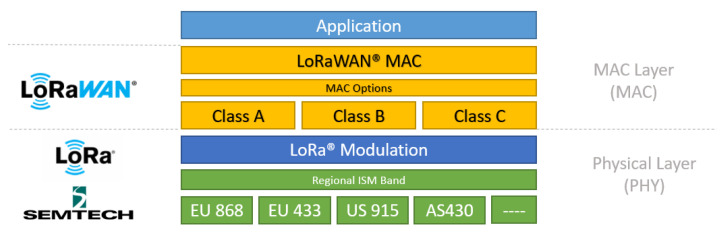
The network layers in LoRa technology [25].

**Figure 3 sensors-24-01621-f003:**
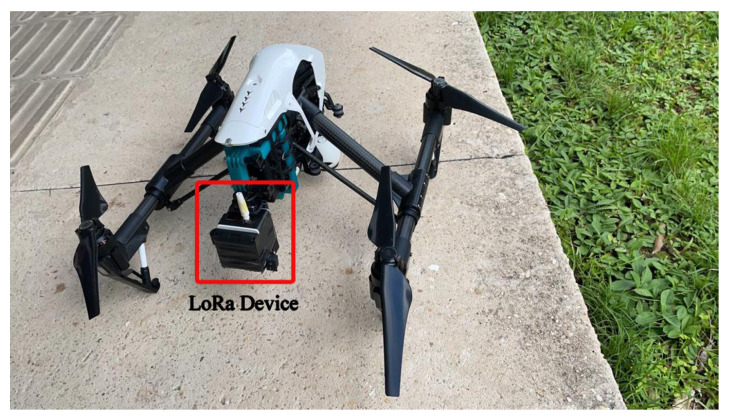
Transmitter equipment: Drone + LoRa end node.

**Figure 4 sensors-24-01621-f004:**
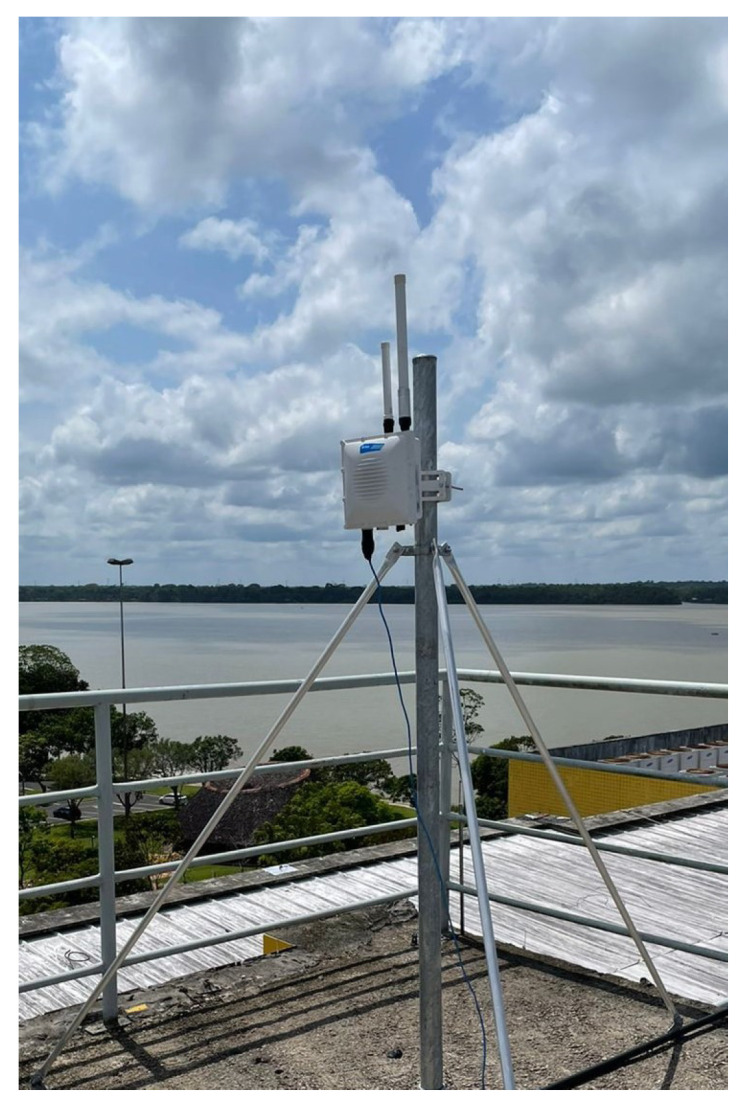
Lora Gateway RAK7249.

**Figure 5 sensors-24-01621-f005:**
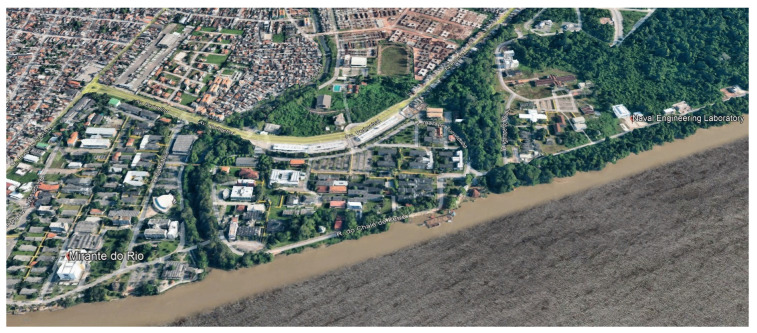
The measurement scenario.

**Figure 6 sensors-24-01621-f006:**
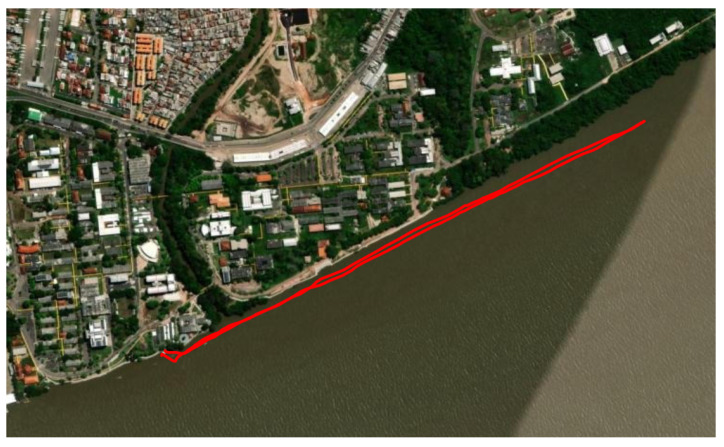
Schematic of the route taken by the drone (route in red).

**Figure 7 sensors-24-01621-f007:**
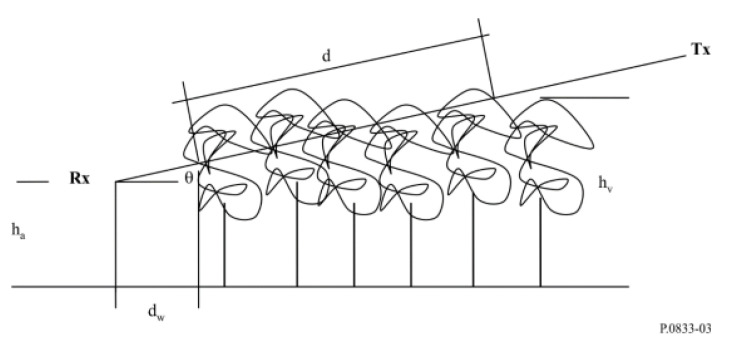
Representative radio path in woodland with vegetation path length (*d*), average tree height ($h_{v}$), height of the Rx antenna over ground ($h_{a}$), radio path elevation (θ), and distance of the antenna to the roadside woodland ($d_{w}$) [49].

**Figure 8 sensors-24-01621-f008:**
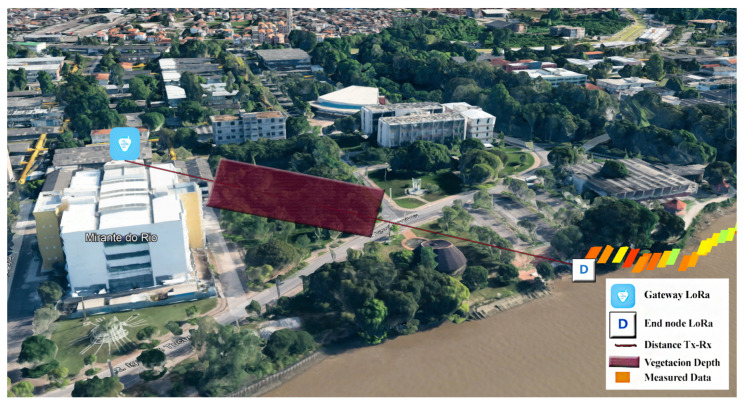
Representation of the location of the gateway (blue icon) and end node (white icon), and the depth of vegetation (red rectangle), used for the calculations of the ITU-R Recommendation P-833 model. The icons in shades of yellow, orange, and green next to the end node icon represent the measured data.

**Figure 9 sensors-24-01621-f009:**
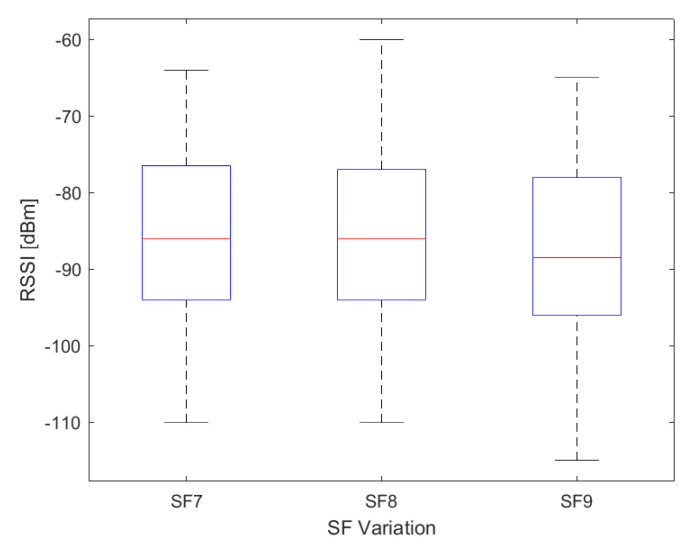
Box plot of the distribution of RSSI data measured, with −86 dBm, −86 dBm, and −88.5 dBm for the median (in red) of the data in SF7, SF8, and SF9.

**Figure 10 sensors-24-01621-f010:**
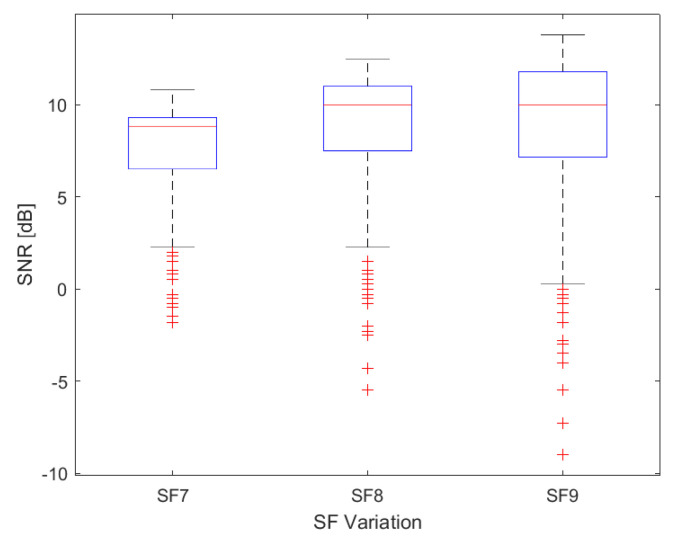
Box plot of the distribution of SNR data measured, with 8.8 dB, 10 dB, and 10 dB for the median (in red) of the data in SF7, SF8, and SF9.

**Figure 11 sensors-24-01621-f011:**
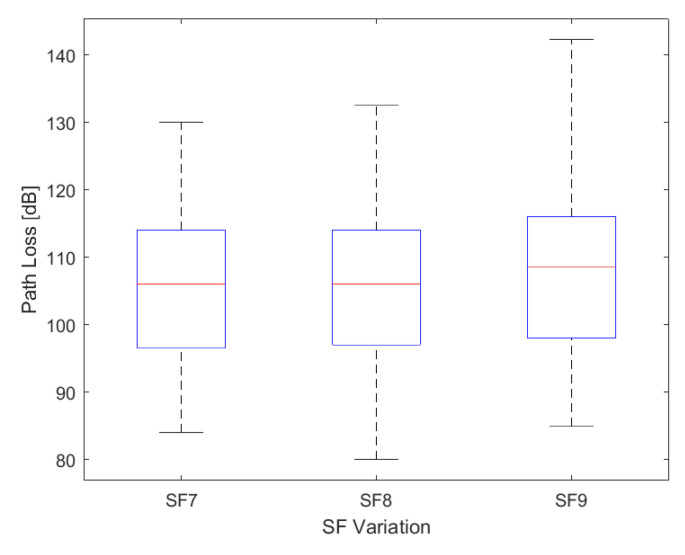
Box plot of the distribution of path loss variation of data measured, with 106 dB, 106 dB, and 108.5 dB for the median (in red) of the data in SF7, SF8, and SF9.

**Figure 12 sensors-24-01621-f012:**
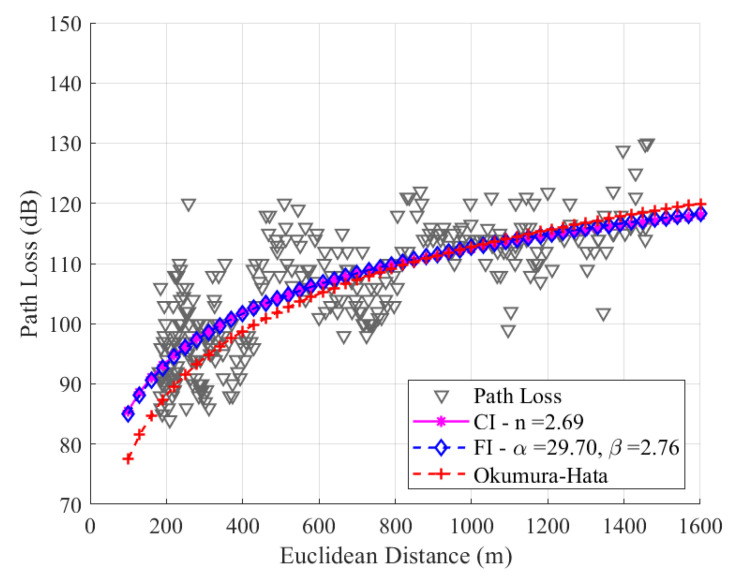
Path loss data and the modeling with CI, FI, and Okumura–Hata models for SF7.

**Figure 13 sensors-24-01621-f013:**
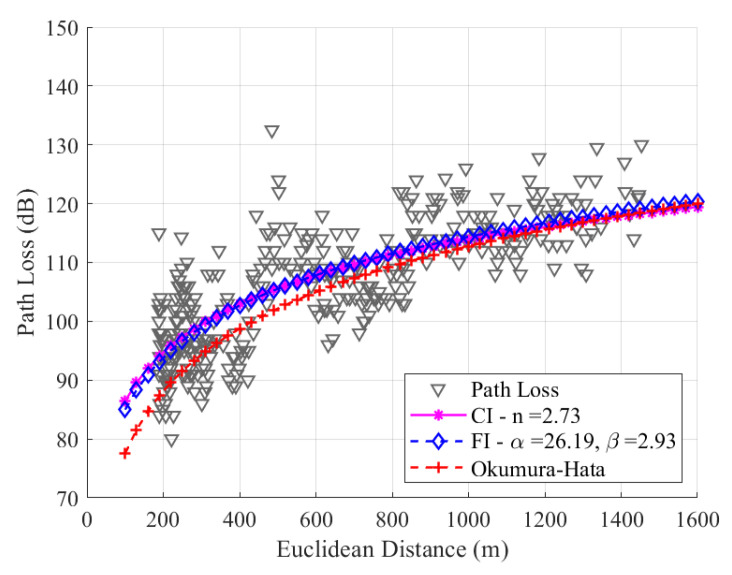
Path loss data and the modeling with CI, FI, and Okumura–Hata models for SF8.

**Figure 14 sensors-24-01621-f014:**
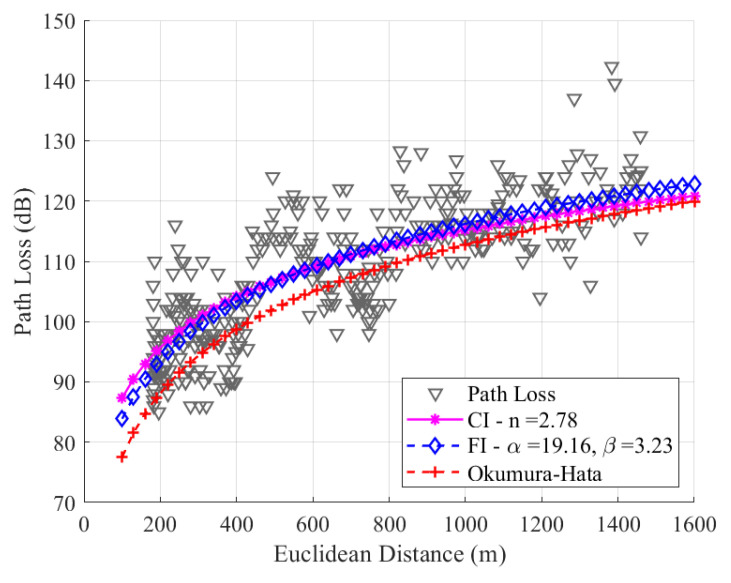
Path loss data and the modeling with CI, FI, and Okumura–Hata models for SF9.

**Figure 15 sensors-24-01621-f015:**
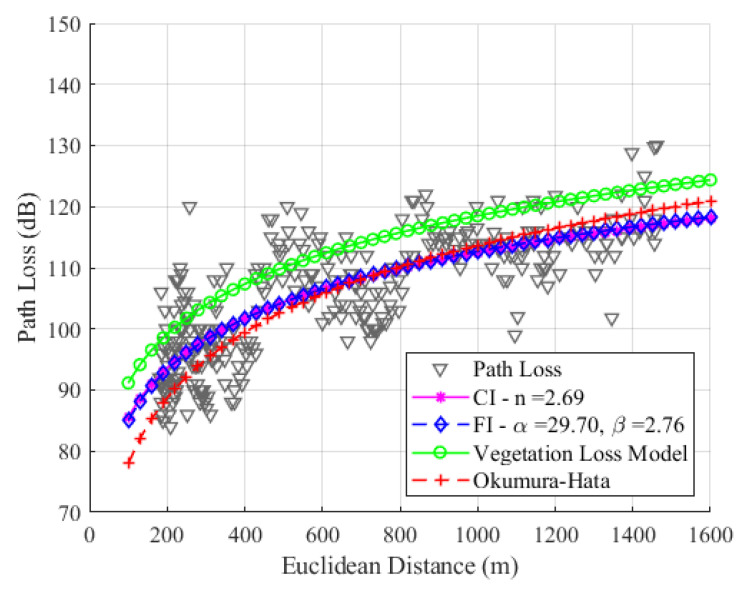
Path loss models for SF7 with CI (magenta), FI (blue), Okumura–Hata (red), and proposed vegetation model (green).

**Figure 16 sensors-24-01621-f016:**
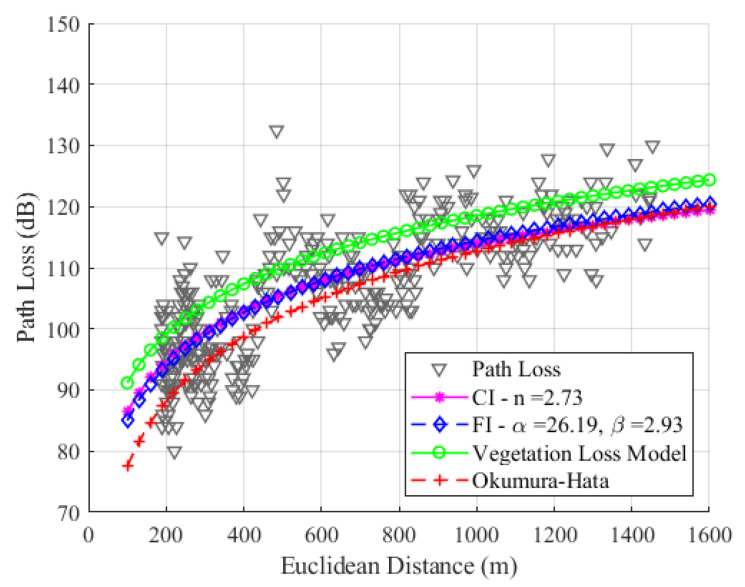
Path loss models for SF8 with CI (magenta), FI (blue), Okumura–Hata (red), and proposed vegetation model (green).

**Figure 17 sensors-24-01621-f017:**
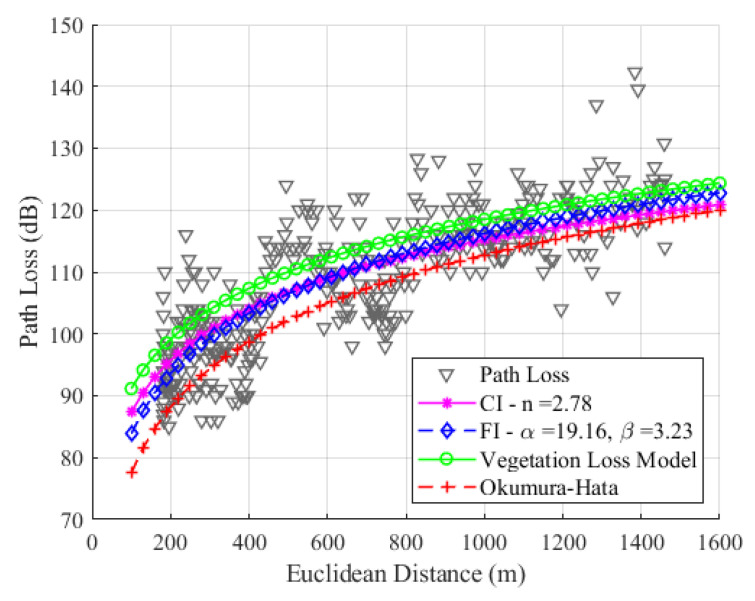
Path loss models for SF9 with CI (magenta), FI (blue), Okumura–Hata (red), and proposed vegetation model (green).

**Figure 18 sensors-24-01621-f018:**
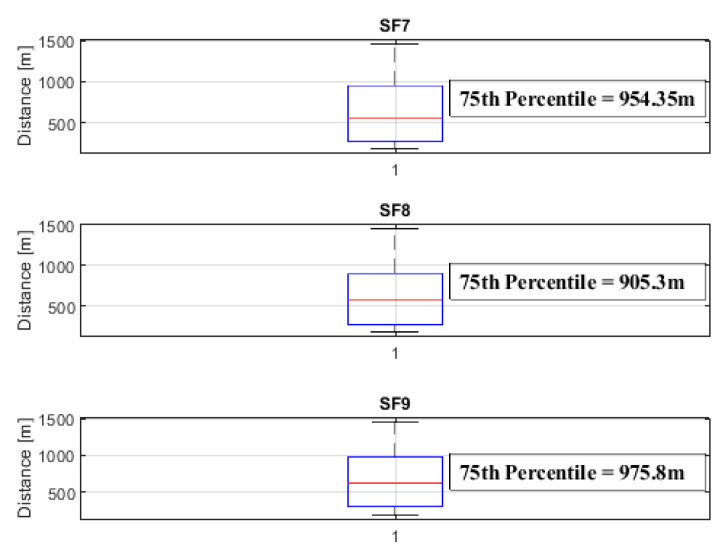
Box plot of the distribution of distance data in SF7, SF8, and SF9 with distance values of 954.35 m, 905.3 m, and 975.8 m representing the values of Q3 (in red) for the three SFs, respectively.

**Table 1 sensors-24-01621-t001:** Related work for LoRA applications over water and in environments with vegetation.

**LoRa Applications over Water**				
**Reference**	**Frequency (MHz)**	**Scenario**	**Coverage (km)**	**Spreading Factor**
[17]	868 (NLOS)	Sea	12.96	7–12
[18]	433 (LOS)	Sea	0.4	7–10
[19]	433/868 (LOS) 868 (NLOS)	Sea	22 28	7/10/12
[24]	915 (LOS)	River	30	7
[27]	433 (NLOS)	Sea	4	12
[28]	868 (LOS)	Sea	4.15	7/9/12
**LoRa Applications with Vegetation**				
**Reference**	**Frequency (MHz)**	**Scenario**	**Coverage (km)**	**Spreading Factor**
[29]	868 (LOS) 868 (NLOS)	Rural with trees	47	12
[30]	915 (NLOS)	Forest	0.25	12
[31]	433 (NLOS)	Forest	X	12
[28]	868 (LOS)	Forest	0.71	7/9/12
[32]	915 (NLOS)	Rural with trees	1.6	7–10
[33]	868 (NLOS)	Forest	1.75	7–12
[34]	868 (NLOS) 915 (NLOS)	Forest Forest near lake	0.23 1	12

**Table 2 sensors-24-01621-t002:** Measurement setup.

Equipment/Parameters	Values/Description
End node	Arduino + LoRa LMIC bib
Gateway	LoRa Gateway RAK 7249
Frequency	915 MHz
Tx Power	20 dBm
Height of gateway	30 m
Height of end node	3 m
Modulation	Chirp spread spectrum
Spreading factor	7/8/9
BW	125 KHz
Code Rate	4/5
Channel	Uplink
Sending message time interval	1 s
GPS module	Quectel L80

**Table 3 sensors-24-01621-t003:** The values of the standard deviation (σ) of the RSSI and SNR values for SF7, SF8, and SF9.

	SF7	SF8	SF9
RSSI	10.06	10.34	10.88
SNR	2.81	3.21	4.06

**Table 4 sensors-24-01621-t004:** Average values of RSSI and SNR and rate of received packets in SF7, SF8, and SF9.

	${\mathrm{RSSI}}_{\mathrm{mean}}$ (dBm)	${\mathrm{SNR}}_{\mathrm{mean}}$ (dB)	RRP (%)
**SF7**	−84.62	7.54	75.78
**SF8**	−85.71	8.71	86.29
**SF9**	−87.76	8.76	92

**Table 5 sensors-24-01621-t005:** Summary of the parameters of channel models for SF7, SF8, and SF9 data.

Spreading Factor	Model	Intercept (dB)	Path Loss Exponent n	RMSE (dB)
SF7	CI	31.36	2.69	6.17
	FI	α = 29.7	β = 2.76	6.17
	Okumura Hata	-	-	6.91
	$PL_{veg}$	31.36	2	8.43
SF8	CI	31.36	2.73	6.57
	FI	α = 26.19	β = 2.93	6.55
	Okumura Hata	-	-	7.53
	$PL_{veg}$	31.36	2	7.98
SF9	CI	31.36	2.78	6.68
	FI	α = 19.16	β = 3.23	6.57
	Okumura Hata	-	-	7.87
	$PL_{veg}$	31.36	2	7.5

## Data Availability

The authors reserve the right not to disclose the private dataset used in this study.

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
