# Peer review of "LoRa Technology Propagation Models for IoT Network Planning in the Amazon Regions"

_sensors, 2024, doi:10.3390/s24051621_

Round 1
Reviewer 1 Report
Comments and Suggestions for Authors
This paper proposes large-scale modeling of propagation loss channels for the LoRaWAN protocol at a frequency of 915 MHz in the context of low-power wide-area networks (LPWANs). The work considers the geographical features of the Amazon Region which make it extremely difficult to design and implement broadcasting and telecommunications networks there. In order to serve riverbank communities with unstable or nonexistent telecommunications systems, the study intends to help with the planning of IoT networks by taking end node mobility into account. The study is based on data measurements to gather data on the geolocated sites, signal-to-noise ratio, and received signal strength indication while taking various spreading factors into account. In uplink, empirical Close-In (CI) and Floating Intercept (FI) propagation models for path loss prediction have been studied.
General Positive Comments:
1- In general, the work of this manuscript is clearly written in its abstract, proposed methodology, measurements, results and comparison.
2- The manuscript is well organized with good comparison study.
3- The references are recent and they are enough for this work.
However, there are some required comments to improve this work as follows:
1- The manuscript title is somehow long including abbreviations such as IoT and it is good to be revised. One suggestion can be for example: Propagation Models of Low-Power Wide Area Networks (LPWANs) for IoT Applications in Amazon Region.
2- The Abstract is good; but some abbreviations are not defined such as Close-In (CI) and Floating Intercept (FI); IoT networks, SF9, etc. Please revise it.
3- In Abstract, the authors did not mention the software tools used in implementation and verifying the models.
4- The keywords are few; please add extra keywords such as LPWAN, path loss etc.
5- What is the software tool used in this work? Please mention this in Abstract.
6- It is good to add extra related work for different LoRA applications, in Table in Section 2 LoRa Overview.
7- Line 428: Subsection 5.2.2 should be changed to avoid the conflict with the caption of Figure 11 5.2.2. Path Loss vs Distance for SF7, SF8, and SF9.
8- How the parameter values of Alfa and Beta in Figures 13-15 are chosen different? Why? Please clarify this in the figures and in Table 4.
9- The Future works mentioned in Conclusions are not clearly described. Further works should be mentioned to show how the results of this study can be used in IoT applications.
10- Please check the definitions of all symbols mentioned in the equations of this work.
Comments on the Quality of English LanguageIt is fine.
Author Response
Dear Reviewer,
Thank you for your recommendations. Please find attached the "Response Letter (Sensors) - Reviewer 1" file with the point-by-point response to your comments.
If you would like access to the other reviewer's letter, I have forwarded a copy of both letters to Ms. Star Chen (Assistant Editor).
Best Regards,
Wirlan Gomes Lima

Reviewer 2 Report
Comments and Suggestions for Authors
See attached file.

Author Response
Dear Reviewer,
Thank you for your recommendations. Please find attached the "Response Letter (Sensors) - Reviewer 2" file with the point-by-point response to your comments.
If you would like access to the other reviewer's letter, I have forwarded a copy of both letters to Ms. Star Chen (Assistant Editor).
Best Regards,
Wirlan Gomes Lima

Round 2
Reviewer 2 Report
Comments and Suggestions for Authors
All of my comments were properly addressed. Nice work. Well done!